# Clinical Consequences and Functional Impact of the Rare S737F CFTR Variant and Its Responsiveness to CFTR Modulators

**DOI:** 10.3390/ijms24076576

**Published:** 2023-03-31

**Authors:** Vito Terlizzi, Emanuela Pesce, Valeria Capurro, Valeria Tomati, Mariateresa Lena, Cristina Pastorino, Renata Bocciardi, Federico Zara, Claudia Centrone, Giovanni Taccetti, Carlo Castellani, Nicoletta Pedemonte

**Affiliations:** 1Department of Paediatric Medicine, Meyer Children’s Hospital IRCCS, Cystic Fibrosis Regional Reference Center, 50139 Florence, Italy; 2UOC Genetica Medica, IRCCS Istituto Giannina Gaslini, 16147 Genova, Italy; 3Department of Neurosciences, Rehabilitation, Ophthalmology, Genetics, Maternal and Child Health (DINOGMI), University of Genoa, 16126 Genova, Italy; 4Diagnostic Genetics Unit, Careggi University Hospital, 50134 Florence, Italy; 5UOSD Centro Fibrosi Cistica, IRCCS Istituto Giannina Gaslini, 16147 Genova, Italy

**Keywords:** nasal primary cells cultures, elexacaftor, tezacaftor, ivacaftor, CRMS, CFSPID

## Abstract

S737F is a Cystic Fibrosis (CF) transmembrane conductance regulator (CFTR) missense variant. The aim of our study was to describe the clinical features of a cohort of individuals carrying this variant. In parallel, by exploiting ex vivo functional and molecular analyses on nasal epithelia derived from a subset of S737F carriers, we evaluated its functional impact on CFTR protein as well as its responsiveness to CFTR modulators. We retrospectively collected clinical data of all individuals bearing at least one S737F CFTR variant and followed at the CF Centre of Tuscany region (Italy). Nasal brushing was performed in cooperating individuals. At study end clinical data were available for 10 subjects (mean age: 14 years; range 1–44 years; 3 adult individuals). Five asymptomatic subjects had CF, 2 were CRMS/CFSPID and 3 had an inconclusive diagnosis. Ex vivo analysis on nasal epithelia demonstrated different levels of CF activity. In particular, epithelia derived from asymptomatic CF subjects and from one of the subjects with inconclusive diagnosis showed reduced CFTR activity that could be rescued by treatment with CFTR modulators. On the contrary, in the epithelia derived from the other two individuals with an inconclusive diagnosis, the CFTR-mediated current was similar to that observed in epithelia derived from healthy donors. In vitro functional and biochemical analysis on S737F-CFTR expressed in immortalized bronchial cells highlighted a modest impairment of the channel activity, that was improved by treatment with ivacaftor alone or in combination with tezacaftor/elexacaftor. Our study provide evidence towards the evaluation of CFTR function on ex vivo nasal epithelial cell models as a new assay to help clinicians to classify individuals, in presence of discordance between clinical picture, sweat test and genetic profile.

## 1. Introduction

Cystic fibrosis (CF) is a multisystem disease caused by variants causing deficient or dysfunctional CF transmembrane conductance regulator (CFTR) protein. The CF phenotype is characterized by lung disease (bronchiectasis with persistent airway-based infection and inflammation), exocrine pancreatic insufficiency associated with malabsorption contributing to undernutrition, impaired growth, hepatobiliary manifestations, and male infertility [1]. In individuals presenting with a positive newborn bloodspot screening (NBS), i.e., with an immunoreactive trypsinogen above the cut-off, clinical features consistent with CF, or a positive family history, a diagnosis of CF can be made if the sweat chloride (SC) value is ≥60 mmol/L or in presence of 2 CFTR causing variants [2].

More than 2000 CFTR variants have been recorded so far worldwide (http://www.genet.sickkids.on.ca/app, accessed on 26 February 2023) although to date only 401 CFTR variants are known to be CF-causing (https://cftr2.org/, accessed on 26 February 2023) on the basis of functional studies. They are classified into 6 categories, according to their impact on the production, trafficking, functioning or stability of the CFTR channel. Variants belonging to classes I, II and III usually result in little to no CFTR activity, leading to severe clinical outcomes, whilst variants from classes IV, V and VI allow significant residual CFTR function leading to milder phenotypes [1]. CFTR gene sequencing may detect variants lacking a clear and univocal genotype–phenotype correlation [3,4,5], i.e., variants of varying clinical consequence (VVCC), non CF-causing or variants of unknown significance (VUS) (https://cftr2.org/, accessed on 26 February 2023). These findings are a frequent challenge, especially in CFTR-related metabolic syndrome (CRMS), CF screen positive, inconclusive diagnosis (CFSPID) infants, i.e., infants with positive NBS results and inconclusive diagnosis [6], limiting the possibility to provide adequate information to parents.

S737F (c.2210C > T; p.Ser737Phe) is a CFTR missense variant, located in exon 13 and characterized by a substitution of serine with phenylalanine in position 737. Serine 737 is one of the phosphorylation sites in the regulatory domain of CFTR, involved in regulation of CFTR protein function and expression at the plasma membrane [7]. In 2018 we described the clinical features of nine subjects carrying at least one S737F and followed at the Florence CF centre, Italy [8]. This variant was associated to hypochloremic alkalosis in childhood, mild CF phenotype in teenage years and a residual function of CFTR protein, evaluated on human nasal epithelial (HNE) cells obtained by nasal brushing in two CF individuals [8]. Subsequently, S737F was included among the 177 variants that are considered to be responsive in vitro to elexacaftor (ELX)/tezacaftor (TEZ)/ivacaftor (IVA) (https://pi.vrtx.com/files/uspi_elexacaftor_tezacaftor_ivacaftor.pdf, accessed on 26 February 2023) and included in the U.S. Food and Drug Administration (FDA) list of CFTR variants approved for CF treatment. No further data on follow up or functional analysis have since been reported.

In this paper we report the clinical features of a cohort of individuals carrying at least one S737F CFTR variant, as well as the results of the functional studies performed on patient-derived HNE cells and of the characterization of the activity and expression pattern of the S737F variant performed on a heterologous expression system.

## 2. Results

### 2.1. Genotype, Clinical Data and Clinical Course

Thirteen subjects (5 females) carrying at least one S737F CFTR variant were identified, three were lost to follow up, and clinical data were available for 10 subjects (mean age at 31 September 2022: 14 years; range 1–44 years; 3 adult individuals). One of these was homozygous for S737F (Table 1). No complex alleles were found. All but one adult individual had undergone CF NBS, according to the Tuscany region algorithm [9].

At first evaluation:−one had a diagnosis of CF in presence of positive NBS and pathological SC level (66 mEq/L);−six were diagnosed with CRMS/CFSPID on the grounds of SC in intermediate or normal range and another CFTR variant on the second allele;−three were labelled as having an inconclusive diagnosis in presence of negative result to CF NBS, isolated episode of hypochloremic alkalosis and SC testing in the intermediate or pathological range (subject 1 and 3 of Table 1) and no CF related respiratory symptoms and pathological SC testing (subject 10 of Table 1).

Demographic and clinical data at first evaluation are reported in Table 1.

All subjects were pancreatic sufficient

After a mean follow up of 9 years (range 11 months–20.5 years) the diagnoses were modified as follows (Table 1):−4 out of 6 asymptomatic CRMS/CFSPID progressed to a CF diagnosis justified by pathological SC values at a mean age of 2.2 years (range 1.2–3.4);−2 asymptomatic subjects maintained the CRMS/CFSPID diagnosis, after a mean follow up of 18 months, in presence of SC value in normal or intermediate range;−3 subjects still had an inconclusive diagnosis, in presence of pathological SC values but neither additional symptoms nor bronchiectasis at chest CT scan.

All subjects were still pancreatic sufficient with good nutritional status; lung disease, if any, was mild with FEV1 in normal range (>90%) and in absence of bronchiectasis at chest CT scan for 7 subjects aged over 6 years (Table 1).

None of the 3 adult subjects had CBAVD.

No subjects required during follow up oral or intravenous antibiotic therapy for PEx.

Finally, no subjects had chronic Pa infection.

### 2.2. Functional Evaluation of CFTR Activity in Patient-Derived Nasal Epithelial Cells

In order to evaluate the potential correlation between clinical features and CFTR activity, we exploited ex vivo nasal epithelia cell models as a tool to directly evaluate variant CFTR activity and its pharmacological response in patient-derived cells by means of electrophysiological techniques. In addition, we expressed the S737F-CFTR in a heterologous expression system (the immortalized bronchial CFBE41o- cell line) to perform an in-depth characterization of the impact of the S737F variant on CFTR channel protein activity and expression using functional and biochemical assays. To gain further knowledge on the processing and activity of the S737F variant, we exploited also CFTR modulators as research tools to unmask possible defects that could not be easily detected in the presence of a marked residual function.

Nasal cells derived from nasal brushings of subjects 1-3-4-5-10 (initially having either an inconclusive diagnosis or a CRMS/CFSPID label; see Table 1) were cultured as described in the Section 4, plated onto permeable supports, and differentiated into epithelia under air-liquid condition for 18 days. Nasal epithelia were treated for 24 h with CFTR modulators added in the basolateral medium. Nasal epithelia were then mounted in a perfusion chamber and short-circuit current measurements were performed to determine CFTR-dependent Cl- secretion (Figure 1). During the recordings, amiloride (10 µM) was applied to inhibit the epithelial sodium channel ENaC, then the membrane-permeable cAMP analogue CPT-cAMP (100 µM) was added to increase intracellular cAMP level (thus mimicking physiological stimulation). CFTR currents were then inhibited by adding the CFTR inhibitor-172 (inh-172, 20 µM) (Figure 1). To evaluate rescue by potentiator, IVA (1 µM) was applied after CPT-cAMP and then the current was inhibited by inh-172. Total CFTR activity was estimated, for each epithelium, as the amplitude of the current drop caused by the addition of inh-172.

We first examined epithelia derived from subject 1, homozygous for the S737F variant. In epithelia treated for 24 h with vehicle alone (0.18% DMSO), the total CFTR activity elicited by CPT-cAMP was quite large: 24.2 ± 3.6 µA (mean ± SD, *n* = 4). Further stimulation with IVA caused no significant increase of total CFTR activity (26.9 ± 3.6 µA; Figure 1A). We then evaluated CFTR activity in epithelia treated for 24 h with combined ELX/TEZ and stimulated with CPT-cAMP followed by IVA. Also in this case, we did not observe any significant increase in CFTR activity (30.7 ± 2.6 µA; Figure 1A). As comparison, in epithelia derived from healthy subjects, cultured, differentiated, and analyzed under the same conditions used in the present study, the total CFTR-mediated current was approximately 30 µA/cm^2^, as shown in Figure 2 and as recently reported by our group [10]. As a further control, we also recruited two subjects homozygous for the F508del variant. Nasal epithelia derived from these subjects displayed a negligible CFTR-dependent activity (Figure 2B). However, when the epithelia were treated for 24 h with combined ELX/TEZ and stimulated with CPT-cAMP followed by IVA, the CFTR-mediated current increased up to 15–18 µA corresponding approximately to 60% of the activity observed in epithelia derived from healthy subjects (Figure 2B). These results are in agreement with previous data reported by our and other groups [11,12,13].

We then evaluated CFTR-mediate transepithelial chloride secretion in the epithelia of the three individuals compound heterozygous for the S737F variant and a CF-causing variant resembling a non-rescuable, null allele. Nasal epithelia derived from subject 3, compound heterozygous for the S737F and 541delC variants, treated for 24 h with DMSO, displayed reduced CFTR activity (10.0 ± 2.0 µA), that was partially but significantly augmented by stimulation with ivacaftor (12.7 ± 1.1 µA; Figure 1B). When epithelia were pre-treated for 24 h with combined ELX/TEZ, the total CFTR activity elicited by CPT-cAMP plus ivacaftor further increased (16.6 ± 1.8 µA; Figure 1B).

Similarly, vehicle-treated nasal epithelia derived from subject 4, compound heterozygous for the S737F and dele22-24 variants, showed reduced CFTR activity (10.1 ± 1.1 µA), and significant rescue by ivacaftor (15.9 ± 0.7 µA). Similar activity was observed in epithelia pre-treated with ELX/TEZ (18.1 ± 2.0 µA; Figure 1C).

Vehicle-treated epithelia derived from subject 5 (S737F/W1282X), displayed reduced CFTR activity (7.4 ± 1.3 µA), with significant rescue by ivacaftor (12.0 ± 1.1 µA) further improved by pre-treatment with ELX/TEZ (14.8 ± 1.6 µA; Figure 1D).

We then evaluated CFTR activity in nasal epithelia derived from subject 10, heterozygous for the S737F variant. DMSO-treated epithelia showed a large CFTR current (20.4 ± 3.8 µA) with no further improvement upon stimulation with ivacaftor (21.5 ± 4.1 µA) or pre-treatment with ELX/TEZ (26.2 ± 3.5 µA; Figure 1E).

We further investigated in vitro the functional defects and the pharmacological responsiveness to CFTR modulators of the S737F-CFTR variant. To this aim, we expressed the variant protein into a cell model widely used for studies on CFTR pharmacology and biology, the immortalized bronchial epithelial CFBE41o- cell line, stably co-expressing the halide-sensitive Yellow Fluorescent Protein (YFP), that allows to estimate CFTR activity as a function of the YFP quenching rate upon addition of an iodide-containing solution (Figure 3A). As additional controls, we also expressed F508del- and wild-type (wt) CFTR proteins. Under these experimental condition, F508del-CFTR showed a marked response to CFTR-modulating drugs, particularly to the corrector combination ELX/TEZ plus the potentiator ivacaftor, with an overall rescue of the CFTR activity that increased up to approximately 65% that measured in cells transfected with wt-CFTR (Figure 3A,B). S737F-CFTR variant displayed a marked residual activity upon stimulation (approximately 60% of the activity of wt-CFTR) with a cAMP agonist (10 µM forskolin) that was further augmented by ivacaftor addition (up to 65% of the activity of wt-CFTR) (Figure 3A,B). A similar value of CFTR activity was observed upon pre-treatment with ELX/TEZ and stimulation with the cAMP agonist, while when ivacaftor was added, CFTR activity increased up to 75% that of wt-CFTR (Figure 3A,B).

Finally, we confirmed these results by evaluating biochemically the expression pattern of CFTR protein in CFBE41o- cells transiently transfected with F508del-, S737F-, or wt-CFTR (Figure 3C). The mature, fully glycosylated CFTR form (band C) was the most represented in cells expressing the wt-CFTR protein (Figure 3C,D). In cells transfected with F508del CFTR, treated with DMSO, the immature, core glycosylated CFTR protein (band B) was the prevalent form (Figure 3C,D). Upon treatment with ELX/TEZ, the mature CFTR form could be detected, while treatment with the triple ELX/TEZ/IVA combination resulted in a reduced rescue of mature CFTR expression (Figure 3C,D). Cells transfected with S737F-CFTR expressed both CFTR forms, with the mature one being the most represented (Figure 3C,D). Treatment with ELX/TEZ did not alter the expression of both the mature and the immature forms (Figure 3C,D). Similar results were obtained when ivacaftor was administered together with the correctors (ELX/TEZ/IVA treatment) (Figure 3C,D).

## 3. Discussion

This paper provides cumulative data on: (1) the clinical course of a cohort of subjects carrying S737F CFTR variant, including the first reported homozygous subject; (2) their chloride secretion and S737F responsiveness to CFTR-modulating drugs in patient-derived nasal epithelia; (3) function and expression pattern of S737F expressed in a heterologous expression system.

Our data confirm over a longer follow up the mild clinical course of these individuals. All had pancreatic sufficiency with normal lung function, and no bronchiectasis at chest CT. Furthermore, no CF related diseases were found, and the adult males were not azoospermic. In spite of the reassuring clinical situation, we confirm a sweat test trend towards the pathological range [8]. CRMS/CFSPID progressed to CF at a mean age of 2.2 years, according to previous data [6,14] and in presence of a CFTR causing variant on the second allele. Given the young age we cannot exclude a similar course over time also for the child with F508del/S737F genotype (subject 6 in Table 1), still with inconclusive diagnosis at the age of one year. On the other hand, the evolution to CF seems less probable in the other subject with a VVCC on the second allele (G1069R, subject 7 in Table 1) [15,16]. These data will provide adequate information and improve communication with the parents of these children.

Nasal epithelia derived from a S737F homozygous subject displayed a total CFTR-mediated current comparable to that observed in epithelia from healthy subjects, without any significant improvements following treatment with CFTR modulators. Notwithstanding such result, at the end of the clinical study this individual received an inconclusive diagnosis label.

On the contrary, some degree of CFTR dysfunction was observed when we evaluated chloride secretion in epithelia derived from three subjects compound heterozygous for the S737F variant and a CF-causing variant resembling a null allele, not rescuable by CFTR modulators. Indeed, these individuals represent the ideal system to study the function of the variant S737F protein produced by a single allele in the native cell context. Interestingly, epithelia derived from these three individuals displayed reduced CFTR-mediated current, that was significantly increased by ivacaftor. CFTR activity was also improved by pre-treatment with correctors (ELX/TEZ). The residual CFTR-mediated current was approx. 25–30% that observed in epithelia derived from healthy subjects (being approx. 30 µA/cm^2^, [10]. Interestingly, two of the three subjects received a CF diagnosis at the end of the clinical study, while the third received an inconclusive diagnosis label.

The analyses of the S737F-CFTR expressed in immortalized bronchial cells confirmed that this variant is characterized by a decreased activity that can be restored by potentiators. Interestingly, also in this case we observed a slight improvement in CFTR function when cells were pre-treated with correctors, but the biochemical evaluation of CFTR expression pattern did not evidence defective protein maturation. Taken together, the results of our ex vivo and in vitro analyses suggest that S737F-CFTR has a defective activity, that can be rescued by stimulation with a potentiator, while the small increase in CFTR activity upon treatment with correctors seems not per se sufficient to demonstrate defective folding, processing, or trafficking to the plasma membrane of the variant CFTR protein. It is interesting to note that the decreased activity of the S737F variant could be detected (and rescued by modulators) only in patient-derived nasal epithelia derived from subjects compound heterozygous for the S737F variant and a CF-causing variant resembling a null allele. On the contrary, and as said before, in epithelia derived from a S737F homozygous subject detected CFTR activity was in the normality range with no significant increase upon treatment with modulators. This suggests that multiple factors may play a role in the manifestation of CFTR dysfunction, allowing a robust detection of the functional defect only when it exceeds a certain threshold. In addition to this, we should also consider possible effects on the total CFTR expression levels that are known to show interindividual differences. In this regard, although we did not assess total CFTR expression levels for each individual, the reproducibility of our results obtained on epithelia derived from non-CF controls and F508del/F508del donors suggest that eventual differences are unlikely to explain the differences in CFTR activity detected in our cohort.

Three subjects did not meet criteria to support a CF or CFTR-RD diagnosis [2,17]. Their inconclusive diagnosis did not change during follow up. As recently suggested, “CFTR dysfunction compatible with CFTR-RD is defined by evidence of in vivo or ex vivo CFTR dysfunction in the CFTR-RD range in at least 2 different CFTR functional tests [sweat test, nasal potential difference (NPD) and intestinal current measurement (ICM)]” [18,19]. Unfortunately, NPD and ICM are not so easily accessible. In vitro and ex vivo studies performed on HNE cells can be a non-invasive tool that may contribute to classify these individuals, particularly in cases where discordance between clinical picture, SCL and genetics occur [20].

In our cohort the individual compound heterozygous for the S737F and dele22-24 variant, showed a reduced CFTR activity (10.1 ± 1.1 µA), which points towards a diagnosis of CFTR-RD, given that functional threshold for CFTR-RD seems to be between 10% and 30% of normal [21,22]. Higher total CFTR activity was found for the other 2 subjects, homozygous for the S737F variant (24.2 ± 3.6 µA) and heterozygous for the S737F (20.4 ± 3.8 µA). Based on these data we will differentiate the follow up of the latter two, avoiding hyper medicalization.

Our data show that individuals with S737F variant may progress to CF during follow-up, with mild lung disease and insufficient evidence of CF related disorders, even in adult age. The evaluation of CFTR function on ex vivo nasal epithelial cell models can help clinicians to classify individuals, in presence of discordance between clinical picture, sweat test and genetic profile.

## 4. Methods

### 4.1. Patients Population

We performed a retrospective analysis of clinical records of all individuals either homozygous or compound heterozygous for the S737F CFTR variant and followed at the CF Centre of Tuscany region (Italy), until 31 September 2022. Demographic, clinical, genetic and biochemical data regarding all enrolled individuals were extracted from local electronic health records. All subjects had given consent to the recording of their clinical data and their anonymous use for scientific purposes, including descriptive studies. Furthermore, the approval of Ethics Committee was obtained for sampling of HNE cells and for the functional analysis (Florence, Ethics Clearance number 74/2020).

Subjects were labelled as CRMS/CFSPID, CF, and CFTR-related disorder (CFTR-RD) according to guidelines of the CF Foundation [2,23] and European recommendations [17]. All individuals enrolled had all CFTR exons, intronic adjacent regions and proximal 5′-and 3′ -UTR studied by Sanger sequencing or next generation sequencing.

Sweat test was performed according to international guidelines [24]. Pancreatic sufficiency was defined on the basis of at least two values of fecal pancreatic elastase higher than 200 μg/g measured outside acute gastrointestinal diseases [25].

At least two measurements of forced expiratory volume in the 1st second (FEV1) were recorded for subjects aged over 6 years. FEV1 was expressed as a percentage of the predicted value for age and sex according to standardized reference equations for spirometry [26]. A chest computed tomography (CT) scan was performed in individuals over 6 years of age.

*Pseudomonas aeruginosa* (Pa) chronic infection was defined using the modified Leeds criteria [27].

Pulmonary exacerbations (PEx) were defined according to the US CF Foundation criteria [28].

A spermiogram was performed in adult males to exclude the presence of congenital bilateral absence of the vas deferens (CBAVD).

In order to better characterize the functional impact of the S737F variant on CFTR channel and evaluate the response to CFTR modulator drugs, we performed a nasal brushing in cooperating individuals without the F508del variant. Indeed, in the presence of the F508del, it would be difficult to assess the relative contribution of each variant to the total CFTR current measured in patient-derived nasal epithelia.

### 4.2. Primary Nasal Epithelial Cell Culture

Isolation, culture, and differentiation of primary HNE cells were performed as previously described [11]. In brief, HNE cells obtained through a nasal brushing of both nostrils of subjects 1 (donor ID: FI208 of the nasal cell registry), 3 (donor ID: FI046 of the nasal cell registry), 4 (donor ID: FI125 of the nasal cell registry), 5 (donor ID: FI062 of the nasal cell registry), and 10 (donor ID: FI209 of the nasal cell registry). In addition, we also utilized HNE cells previously obtained from three non-CF donors (donor IDs: Ctr178, Ctr153, and Ctr147 of the nasal cell registry) and from two F508del/F508del CF individuals (donor IDs: TT005 and AN232 of the nasal cell registry). HNE cells were cultured and expanded in a serum-free medium (LHC9 mixed with RPMI 1640, 1:1) containing various hormones and supplements, including ROCK and SMAD inhibitors (DMH-1, A-83-01, and Y-27632 compounds) [29]. In the first days, the culture medium was supplemented with a mixture of different antibiotics (including colistin, piperacillin, and tazobactam) to eradicate bacterial contamination. To obtain differentiated epithelia, nasal cells were seeded at high density (500,000 cells/cm^2^) on porous membranes (Snapwell inserts, code 3801, Corning Life Sciences, Corning, NY, USA). After 24 h, the serum-free medium was removed and replaced with Pneumacult ALI medium (StemCell Technologies, Vancouver, BC, Canada) on the basolateral side only. Epithelia differentiation (up to 16–18 days) was performed in air-liquid interface (ALI) condition.

The collection of HNE cells and their study to investigate the mechanisms of transepithelial ion transport were specifically approved by the Ethics Committee of the Istituto Giannina Gaslini following the guidelines of the Italian Ministry of Health (registration number: CER 28/2020, 4 April 2020).

### 4.3. Short-Circuit Current Recordings

Snapwell inserts containing nasal epithelia were mounted in a vertical diffusion chamber resembling a Ussing chamber with internal fluid circulation. Both apical and basolateral hemichambers were filled with 5 mL of a solution containing (in mM) 126 NaCl, 0.38 KH_2_PO_4_, 2.13 K_2_HPO_4_, 1 MgSO_4_, 1 CaCl_2_, 24 NaHCO_3_, and 10 glucose. Both sides were continuously bubbled with a 5% CO_2_–95% air mixture and the temperature of the solution was kept at 37 °C. The transepithelial voltage was short-circuited with a voltage-clamp (DVC-1000, World Precision Instruments, Sarasota, FL, USA; VCC MC8 Physiologic Instruments, Reno, NV, USA) connected to the apical and basolateral chambers via Ag/AgCl electrodes and agar bridges (1 M KCl in 2% agar). The transepithelial voltage was clamped at 0 mV after correcting voltage offsets and fluid resistance compensation. The short-circuit current was recorded by analogical to digital conversion on a personal computer.

### 4.4. Cell Culture

CFBE41o- cells stably expressing the halide-sensitive Yellow Fluorescent Protein (YFP) were generated as previously described [30]. CFBE41o- cells were grown in MEM medium supplemented with 10% FBS, 2 mM L-glutamine, 100 U/mL penicillin, and 100 µg/mL streptomycin (Euroclone, Milano, Italy). For the YFP-based assays of CFTR activity, CFBE41o- cells were plated (50,000 cells/well) on clear-bottom 96-well black microplates (Corning Life Sciences, Corning, NY, USA).

### 4.5. Chemicals, Antibodies and Vectors

The CFTR modulators ivacaftor and tezacaftor were from TargetMol (catalog ID: T2588 and T2263, respectively; Wellesley Hills, MA, USA). Elexacaftor was purchased from MedChemExpress (catalog ID: HY-111772; Monmouth Junction, NJ, USA). The final working concentration used for the CFTR modulators were as follows: elexacaftor, 3 µM; tezacaftor, 10 µM; ivacaftor, 1 µM (when applied acutely during short-circuit current measurements or for the YFP assay) or 5 µM (for 24 h treatment).

The following antibodies were used: mouse monoclonal anti-CFTR (ab769, J.R. Riordan, University of North Carolina at Chapel Hill, and Cystic Fibrosis Foundation Therapeutics); mouse monoclonal anti-GAPDH (sc-32233; Santa Cruz Biotechnology, Inc.; RRID: AB_627679, Dallas, TX, USA); horseradish peroxidase (HRP)-conjugated anti-mouse IgG (ab97023; Abcam; RRID: AB_10679675, Cambridge, UK) and horseradish peroxidase (HRP)-conjugated goat anti-rabbit IgG (0031460; ThermoFisher Scientific, Waltham, MA, USA). Vectors encoding WT-, S737F-, and F508del-CFTR variants were purchased from VectorBuilder (vector IDs available upon request; Neu-Isenburg, Germany).

### 4.6. Transient Transfection of CFBE41o-Cells

The microfluorimetric YFP-based assay and western blot CFTR analysis on CFBE41o- cells expressing the halide-sensitive YFP were performed as previously described [11]. For YFP assay cells were reverse-transfected onto 96-well plates with 0.2 µg per well of the indicated vectors (see Section 4.5). To measure CFTR expression by Western blot, cells were reverse-transfected onto 6-well plates with 2 µg of the indicated vectors (see Section 4.5). Lipofectamine 2000 (ThermoFisher Scientific, Waltham, MA, USA) was used as a transfection agent. Cells were transfected, in Opti- MEM™ Reduced Serum Medium (ThermoFisher Scientific, Waltham, MA, USA). After 6 h, Opti-MEM was carefully replaced with culture medium without antibiotics. A total of 24 h after transfection and plating, cells were treated with correctors or vehicle alone (DMSO) at the indicated concentrations and incubated at 37 °C for an additional 24 h, prior to proceeding with the functional YFP-based assay or to the cell lysis.

### 4.7. YFP-Based Assay for CFTR Activity

CFTR activity was determined by the YFP microfluorimetric assay (details can be found in previous studies [12,31]. Briefly, prior to the assay, cells were washed with PBS (137 mM NaCl, 2.7 mM KCl, 8.1 mM Na_2_HPO_4_, 1.5 mM KH_2_PO_4_, 1 mM CaCl_2_, and 0.5 mM MgCl_2_) and then incubated for 25 min with 60 µL of PBS plus forskolin (20 µM) and IVA (1 µM) at 37 °C, to maximally stimulate the CFTR channel. Cells were then transferred to a microplate reader (FluoStar Galaxy or Fluostar Optima; BMG Labtech, Offenburg, Germany), equipped with high-quality excitation (HQ500/20X: 500 ± 10 nm) and emission (HQ535/30M: 535 ± 15 nm) filters for YFP (Chroma Technology, Bellows Falls, VT, USA). Each assay consisted of a continuous 14-s YFP fluorescence recording with 2 s before and 12 s after injection of 165 µL of an iodide-containing solution (PBS with Cl^−^ replaced by I^−^; final I^−^ concentration 100 mM). Data were normalized to the initial background-subtracted fluorescence. To determine the I^−^ influx rate, the final 11 s of the data for each well were fitted with an exponential function to extrapolate the initial slope (dF/dt).

### 4.8. Western Blot

After transfection, CFBE41o- cells were grown to confluence onto a 6-well plate. The day of cell lysis, cells were washed with ice-cold D-PBS without Ca^2+^/Mg^2+^ and then lysed in RIPA buffer containing a complete protease inhibitor cocktail (Roche, Rotkreuz, Switzerland). Cell lysates were then processed as previously described [11]. In brief, lysates were separated by centrifugation at 15,000× *g* at 4 °C for 10 min. The supernatant protein concentration was calculated using a BCA assay (ThermoFisher Scientific, Waltham, MA, USA) following the manufacturer’s instructions. Proteins (25 µg) were separated onto gradient 4–15% Criterion TGX Precast gels (Bio-rad Laboratories Inc., Hercules, CA, USA), transferred to a nitrocellulose membrane with a Trans-Blot Turbo system (Bio-rad Laboratories Inc., Hercules, CA, USA) and analyzed by Western blotting. CFTR and other proteins of interest were detected using antibodies indicated in the dedicated Section 4 and subsequently visualized by chemiluminescence using the SuperSignalWest Femto or West Dura Substrate (ThermoFisher Scientific, Waltham, MA, USA). Molecular Imager ChemiDoc XRS System (Bio-rad Laboratories Inc., Hercules, CA, USA) was used to monitor the chemiluminescence. Images were analyzed with ImageJ software (National Institutes of Health, Bethesda, MD, USA). Bands were analyzed as region-of-interest (ROI), normalized against the GAPDH loading control.

### 4.9. Statistics

The assumption of normality of data distribution was assessed by applying the Kolmogorov–Smirnov test. An analysis of variance (ANOVA) followed by a post-hoc test was used to avoid “multiple comparisons error”. A parametric ANOVA was performed for normally distributed quantitative variables. A parametric ANOVA followed by the Dunnet multiple comparisons test (all groups against the control group) as a post-hoc test was used to assess statistical significance of the effect of single drug treatments on CFTR activity in CFBE41o- or HNE cells. An ANOVA followed by the Tukey test (for multiple comparisons) as a post-hoc test was instead used in the case of combinations of drugs. When comparing selected pairs of treatment, the statistical significance was tested by ANOVA followed by Bonferroni as a post-hoc test. Normally distributed data are expressed as the mean ± SD and significances are two-sided. Differences were considered statistically significant when *p* < 0.05.

## Figures and Tables

**Figure 1 ijms-24-06576-f001:**
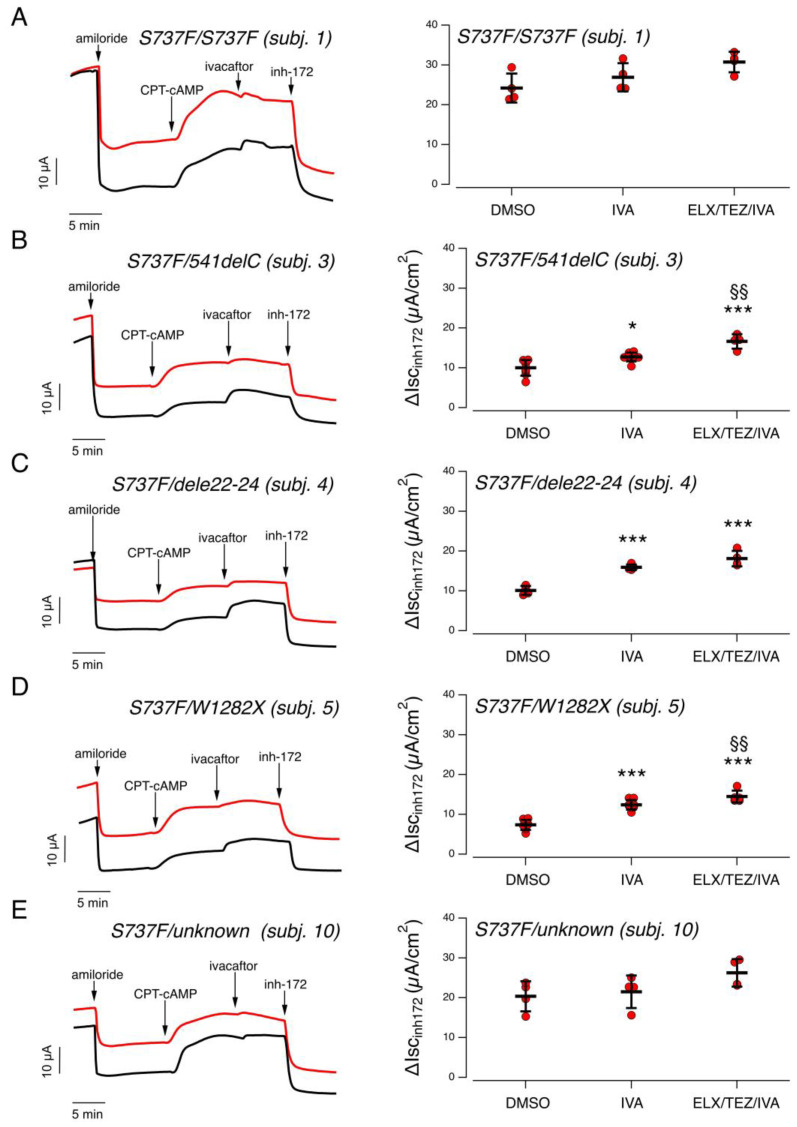
Functional evaluation of modulator treatment on nasal epithelia derived from subjects carrying the S737F variant. (**A**). Left panel: Representative traces of the effect of vehicle alone (DMSO; black trace), or ELX/TEZ (3 µM/10 µM; red trace) on S737F/S737F nasal epithelia (derived from subject 1) with the short-circuit current technique. During the recordings, the epithelia were sequentially treated (as indicated by downward arrows) with amiloride (10 µM), CPT-cAMP (100 µM), ivacaftor (1 µM), and the CFTR inhibitor-172 (inh-172; 20 µM). Right panel: Scatter dot plot showing the summary of results. Data reported are the amplitude of the current blocked by 20 µM inh-172 (ΔIscinh-172) in epithelia treated with vehicle and stimulated with either CPT-cAMP alone (DMSO), or CPT-cAMP plus ivacaftor (IVA) or in epithelia treated with ELX/TEZ and stimulated with CPT-cAMP plus ivacaftor (ELX/TEZ/IVA). (**B**). Representative traces and scatter dot plot summarizing the results obtained on S737F/541delC nasal epithelia (derived from subject 3) of experiments performed as in (**A**). (**C**) Representative traces and scatter dot plot summarizing the results obtained on S737F/dele22-24 nasal epithelia (derived from subject 4) of experiments performed as in (**A**). (**D**). Representative traces and scatter dot plot summarizing the results obtained on S737F/W1282X nasal epithelia (derived from subject 5) of experiments performed as in (**A**). (**E**). Representative traces and scatter dot plot summarizing the results obtained on nasal epithelia (derived from subject 10, S737F heterozygous) of experiments performed as in (**A**). For each donor the number of biological replicates was *n* = 4–8. Symbols indicate statistical significance of treatments: *, *p* < 0.05; ***, *p* < 0.001 vs. DMSO-treated, CPT-cAMP stimulated condition; §§, *p* < 0.01 vs. DMSO-treated, CPT-cAMP + ivacaftor stimulated condition.

**Figure 2 ijms-24-06576-f002:**
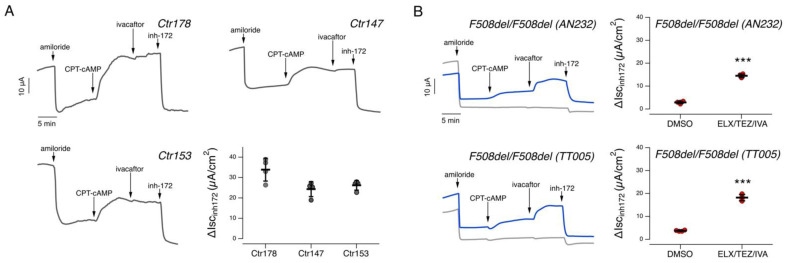
Functional evaluation of modulator treatment on nasal epithelia derived from non-CF subjects or from individuals homozygous for the F508del variant. (**A**). Representative traces recorded on nasal epithelia derived from three non-CF donors with the short-circuit current technique. During the recordings, the epithelia were sequentially treated (as indicated by downward arrows) with amiloride (10 µM), CPT-cAMP (100 µM), ivacaftor (1 µM), and the CFTR inhibitor-172 (inh-172; 20 µM). The scatter dot plot shows the summary of results. Data reported are the amplitude of the current blocked by 20 µM inh-172 (ΔIsc_inh-172_) as measured in epithelia from the different donors. (**B**). Representative traces and scatter dot plots summarizing the results of the effect of vehicle alone (DMSO; gray trace), or ELX/TEZ (3 µM/10 µM; blue trace) obtained on F508del/F508del nasal epithelia (derived from two different individuals) with the short-circuit current technique. During the recordings, the epithelia were sequentially treated as indicated in (**A**). For each donor the number of biological replicates was *n* = 4. Asterisks indicate statistical significance of treatments: ***, *p* < 0.001.

**Figure 3 ijms-24-06576-f003:**
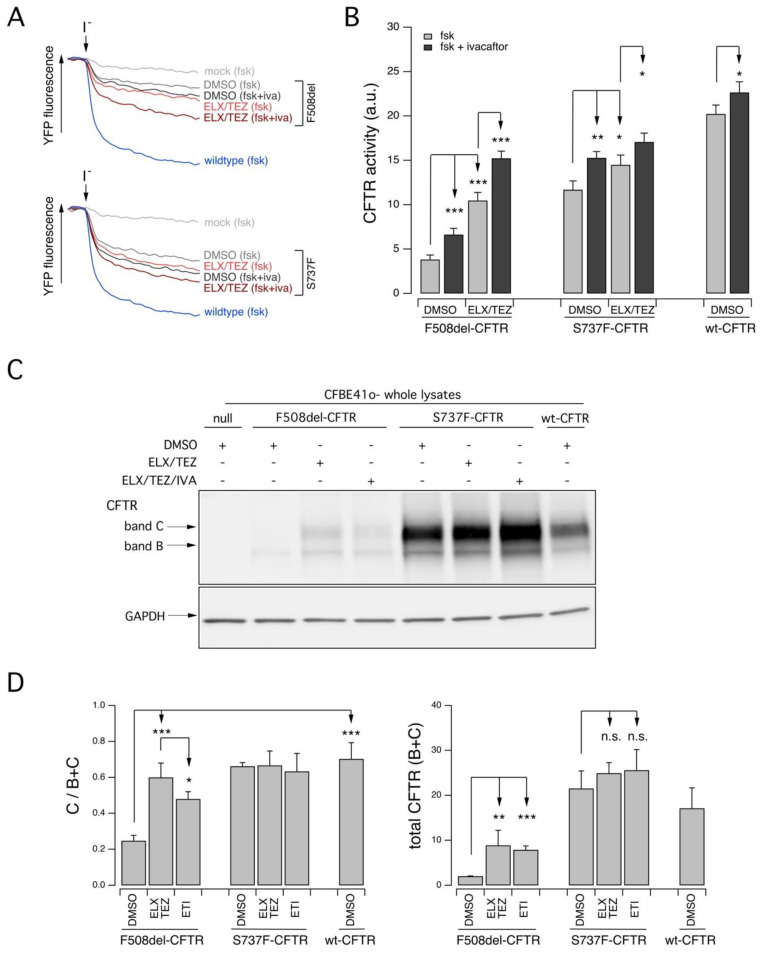
Functional evaluation of the activity and expression pattern of the S737F CFTR variant and of the response to pharmacological treatments on CFBE41o- heterologous expression system. (**A**). Original traces showing the YFP quenching following iodide influx in CFBE41o- cells stably expressing the YFP and transiently transfected with an empty vector (mock) or a vector encoding for wt or variant CFTR proteins (F508del or S747F), treated for 24 h with DMSO alone (vehicle), or (for variant CFTR only) with ELX/TEZ (3 µM/10 µM) and stimulated during the assay with forskolin (20 µM; fsk) or fsk + ivacaftor (1 µM; fsk + iva). (**B**). The bar graph shows the activity of S737F CFTR and, for comparison, of F508del- and wt-CFTR, determined as a function of the YFP quenching rate following iodide influx, obtained from experiments described in A. Data are means ± SD (*n* = 3). (**C**). Biochemical analysis of the S737F-CFTR expression pattern in CFBE41o-cells. The representative western blot image shows CFTR electrophoretic mobility in cell lysates following treatment for 24 h with vehicle alone (DMSO) or Luma (3 µM), or ELX/TEZ (3 µM/10 µM), or ELX/TEZ/IVA (ETI; 3 µM/10 µM/5 µM) prior to lysis. Lysates of parental cells have been included as control for antibody specificity. (**D**). The graphs show quantification of relative CFTR band C expression (C/B + C; left panel) and total CFTR expression (B + C; right panel) obtained by densitometry of the western blot experiments shown in (**C**). Data are means ± SD (*n* = 3). Asterisks indicate statistical significance of treatments: *, *p* < 0.05; **, *p* < 0.01; ***, *p* < 0.001, while n.s. indicates not significant differences.

**Table 1 ijms-24-06576-t001:** Clinical data and diagnosis/label at first evaluation and at end of the study for the enrolled subjects.

Subject	IRT ^ (ng/mL)	Age at First Evaluation (m/y)	Reason for Sweat Testing	Second *CFTR* Variant	First SC Value	Diagnosis/Label at First Evaluation	Age at 31 September 2022 (Years)	Microbiological	Last BMI-Weight for Length Pc ^×^	Last FEV_1_	Last SC Value	Diagnosis/Label at Study End
1 ˜	31	20 m	Hypochloremic alkalosis	S737F	45	Inconclusive diagnosis	14.5	MSSA	21.0	111	98	Inconclusive diagnosis
2	76	1 m	Positive NBS	F508del	35	CRMS/CFSPID	15	No pathogenic bacteria	21.5	108	78	CF
3 ˜	108	1 m	Positive NBS	541delC	48	CRMS/CFSPID	13	MSSA	19.0	107	94	CF
4 ˜^,#^	37	10 m	Hypochloremic alkalosis	22, 23, 24 del	71	Inconclusive diagnosis	20.5	MSSA	21.9	102	89	Inconclusive diagnosis
5 ˜^,#^	64	1 m	Positive NBS	W1282X	51	CRMS/CFSPID	20.8	MSSA	20.5	102	121	CF
6	65	1 m	Positive NBS	F508del	51	CRMS/CFSPID	1.0	No pathogenic bacteria	50°	n.a	48	CRMS/CFSPID
7	61	1 m	Positive NBS	G1069R	25	CRMS/CFSPID	2.3	No pathogenic bacteria	16.0	n.a	20	CRMS/CFSPID
8	48	1 m	Positive NBS	F508del	66	CF	2.5	MSSA	14.0	n.a	66	CF
9	76	1 m	Positive NBS	F508del	51	CRMS/CFSPID	7.0	MSSA	15.4	116	69	CF
10 ˜^,#^	n.a	40 y	Respiratory Symptoms	Exon scanning negative	62	Inconclusive diagnosis	44.0	No pathogenic bacteria	29.0	95	82	Inconclusive diagnosis

^ >99 centile with IRT value in the range 47–51; lost to follow up; ˜ performing nasal brushing; ^#^ these subjects were evaluated for CBAVD and all were found not to be azoospermic; ^×^ percentile weight/length for the child under two years at 31 September 2022. Abbreviations: CFTR: Cystic fibrosis transmembrane conductance regulator; CRMS: CFTR-related metabolic syndrome; CFSPID: CF screen positive, inconclusive diagnosis; CF: Cystic Fibrosis; SC: sweat chloride; IRT: immunoreactive trypsinogen; NBS: newborn bloodspot screening; BMI: body mass index; FEV_1_: forced expiratory volume in the 1st second; MSSA: methicillin-susceptible Staphylococcus aureus; CBAVD: congenital bilateral absence of the vas deferens; n.a.: not applicable.

## Data Availability

All the data are contained within the article.

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
