# Peer review of "Clinical Consequences and Functional Impact of the Rare S737F CFTR Variant and Its Responsiveness to CFTR Modulators"

_ijms, 2023, doi:10.3390/ijms24076576_

Round 1
Reviewer 1 Report
The authors reported the clinical features of S737F CFTR variant and investigated the rescue of its defective CFTR function with CFTR-modulating drugs through the functional analysis using patient-derived nasal epithelial cells. They made a lot of effort to find a cohort of patients carrying S737F CFTR variant and long-term follow up as well as ex vivo and in vivo analyses. Nevertheless, I cannot get the definitive clinical implications from this manuscript
Major comments.
1) This manuscript is diffuse. The conclusion is not in concert with the title. The focus shifts from the S737F CFTR variant to ex vivo nasal epithelial cell models to evaluate CFTR function. But the patients carrying S737F CFTR variant are clinically normal pancreatic and respiratory function, proving the rescue by CFTR-modulating drugs at this point with ex vivo model does not seem to have clinical importance. Rather, focusing S737F CFTR variant, in depth study and discussion depending on the second variants of S737F as well as comparative analysis with the reported results of other common CFTR variants might be more helpful to raise the value of this manuscript.
2) The design and data using ex vivo and in vivo models are not convincing. The effects of rescue by modifiers are not prominent to notice with Fig. 1. To make believe, the authors should add the tracings of clear positive and negative controls (normal and homozygous F508del control), as well as present the tracings from the same patient overlapped together to see the difference clearly and add indicating markers in the tracing used for the conversion into graph. As the way shown in Fig.1 (side-by-side with a certain distance), the readers cannot follow how to convert from the tracings to the graph in Fig.1. Also showing the result from S737F/F508del patient will give an important information. In addition, in Fig 2A, showing the raw data (tracings) of YFP assay for CFTR activity together can make more reliable.
Minor comments.
1) The paragraphs are too short and sometimes fragmentary, especially in result section. Try integrate the paragraphs together.
2) Abbreviations are not organized systematically. Some abbreviations are first used without explaining full names, others are showing repeatedly or wrong (HNE, HS-YFP, VX-770, CMRS etc).
3) In result section, Figures 2A and 2B are repeated referred in the same paragraphs.
4) Some sentences are awkward (22-23,64, 70 , 122, 137) or hard to figure out the meanings (257 BMI-pc weight/length, 343).
Author Response
AUTHORS’ REPLIES TO REVIEWERS’ COMMENTS
Clinical consequences and functional impact of the rare S737F CFTR variant and its responsiveness to CFTR modulators
Terlizzi V et al.
REVIEWER #1
The authors reported the clinical features of S737F CFTR variant and investigated the rescue of its defective CFTR function with CFTR-modulating drugs through the functional analysis using patient-derived nasal epithelial cells. They made a lot of effort to find a cohort of patients carrying S737F CFTR variant and long-term follow up as well as ex vivo and in vivo analyses. Nevertheless, I cannot get the definitive clinical implications from this manuscript.
Re: We thank the reviewer for the comments and for taking the time to review our paper.
Major comments.
1) This manuscript is diffuse. The conclusion is not in concert with the title. The focus shifts from the S737F CFTR variant to ex vivo nasal epithelial cell models to evaluate CFTR function. But the patients carrying S737F CFTR variant are clinically normal pancreatic and respiratory function, proving the rescue by CFTR-modulating drugs at this point with ex vivo model does not seem to have clinical importance. Rather, focusing S737F CFTR variant, in depth study and discussion depending on the second variants of S737F as well as comparative analysis with the reported results of other common CFTR variants might be more helpful to raise the value of this manuscript.
Re: We are aware that the study reports a lot of data. In our opinion the correlation between clinical and functional data is useful and helps to better understand this variant. However, in depth study and discussion depending on the second variant of the S737F subjects will not add informative data on the investigated individuals. Indeed, the subjects that we selected for the functional studies are either homozygous for the S737F variant or compound heterozygous for the S737F variant and a CF-causing variant resembling a null allele, not rescuable by CFTR modulators. These individuals represent the ideal system to study the function of the variant S737F protein produced by a single allele in the native airway cell context.
However, there is also a misunderstanding about the modulator studies that we performed. The reason to use modulators was not to demonstrate any clinical benefit of these drugs on the S373F subjects. We agree that these individuals are clinically normal in terms of pancreatic and respiratory function. On the contrary we exploited CFTR modulators as research tools to unmask possible functional defects that could not be easily detected in the presence of a marked residual function. As discussed in the text, the S373F variant displays a very modest activity defect that can be rescue by stimulation with a potentiator (in this case, ivacaftor). Similarly, pre-treatment with correctors improves CFTR activity, although the experimental data do not support the presence of a maturation defect (again, as already discussed in the text).
Finally, we do not understand how “a comparative analysis with other common CFTR variants might be more helpful to raise the value of this manuscript”. Nevertheless, we have included additional controls (as also explained in the answer to comment #2) to corroborate the overall quality of our studies.
2) The design and data using ex vivo and in vivo models are not convincing. The effects of rescue by modifiers are not prominent to notice with Fig. 1. To make believe, the authors should add the tracings of clear positive and negative controls (normal and homozygous F508del control), as well as present the tracings from the same patient overlapped together to see the difference clearly and add indicating markers in the tracing used for the conversion into graph. As the way shown in Fig.1 (side-by-side with a certain distance), the readers cannot follow how to convert from the tracings to the graph in Fig.1.
Also showing the result from S737F/F508del patient will give an important information. In addition, in Fig 2A, showing the raw data (tracings) of YFP assay for CFTR activity together can make more reliable.
Re: Following reviewer’s criticism, we have prepared – for reviewing purposes only – a modified version of figure 1 in which original short-circuit current recordings performed on epithelia derived from the same donor are shown overlapped together. In this modified figure, we have also indicated the current delta that is used as an estimate for CFTR activity under the different treatments. Thus, it should be now easier to look at the modulators effect on the epithelia derived from the various donors.
To help the reviewer we have also included in our manuscript a new figure (figure 2 in the revised version of the manuscript) that shows original short-circuit current recordings performed on epithelia derived from three non-CF donors (healthy volunteers) and two F508del homozygous individuals, treated with ETI or vehicle alone. The summaries of the results obtained on these subjects are also shown. Hope the reviewer will agree that these data are indeed convincing regarding the reliability of the analyses.
We disagree with the reviewer’s suggestion that “showing the result from S737F/F508del patient will give an important information”. On the contrary, in the presence of the F508del variant – it would be difficult to assess the relative contribution of each variant to the total CFTR current measured in patient-derived nasal epithelia. Thus, the data would not be informative at all to gain further knowledge on the S373F variant.
Minor comments.
1) The paragraphs are too short and sometimes fragmentary, especially in result section. Try integrate the paragraphs together.
Re: We have carefully checked our text to fix fragmentary sentences. However, the use of short sentences with clear statements is a pre-requisite to facilitate comprehension for non-native English speakers.
2) Abbreviations are not organized systematically. Some abbreviations are first used without explaining full names, others are showing repeatedly or wrong (HNE, HS-YFP, VX-770, CMRS etc).
Re: As suggested, we have fixed the abbreviations. However, we could not find any reference to VX-770 in our manuscript.
3) In result section, Figures 2A and 2B are repeated referred in the same paragraphs.
Re: Usually, the references to figure panels are used whenever necessary to explain where data supporting a specific “claim” are shown. Thus, we do not feel it is an issue if reference to figures are repeatedly reported.
4) Some sentences are awkward (22-23,64, 70, 122, 137) or hard to figure out the meanings (257 BMI-pc weight/length, 343).
Re: We have carefully checked our text to fix possible “awkward” sentences. Pc was by percentile. In children < 2 years of age, it is incorrect to use BMI as a growth parameter.
REVIEWER #2
In this manuscript, Terlizzi et al provide a quite interesting report on the CFTR variant S737F pointing out how collaborative efforts bringing together clinical and translational researchers can provide relevant insight into the effects of rare variants and inform on the best therapeutic options.
Re: We thank the reviewer for his/her appreciation of our collaborative efforts.
There are no major concerns regarding the work presented but some aspects could be improved.
- The term “variant” is preferred over “mutation”. Please correct the few times in which “mutation” is still used. The same applies to the term “individuals” which should be used throughout instead of “patients”.
Re: Done as suggested
- The clinical description in the abstract is hard to follow – consider rewriting those sentences to make it more clear.
Re: Done as suggested.
- Tables 1 and 2 are difficult to follow – as they include some repeated data. Please consider merging them into a single table, probably in “landscape” format.
Re: We have merged the two tables in a one table.
- In figure 1, please consider adding the individual numbers (as they are referred to that in the text) – or instead use only the genotypes in the text.
Re: Following reviewer’s suggestion, we have now included the individual numbers in figure 1
- One of the puzzling results is the lack of significant improvement in function after IVA or ETI for the homozygous individual, when compared to the increase observed in individuals in which S737F appears in heterozygosity with a null allele. The authors could probably comment on this in the Discussion and on the possibility that there is a “ceiling” for CFTR activity (?).
Re: Indeed, we have preliminary data suggesting such a hypothesis. However, further evidence is needed before we can propose it. Nevertheless, as proposed, we are now briefly commenting this puzzling result in the Discussion.
- Another aspect that could be mentioned in the Discussion is the total CFTR expression levels in the different individuals – this is not assessed but could contribute to explaining some of the differences.
Re: The reviewer is certainly right. We are now discussing the interindividual differences in CFTR expression levels as proposed.
- Finally, the authors could also check, in the western blot data, if total CFTR (and band C) is increased after ETI treatment. Although processing (C over B+C) is not changed, total amount of mature CFTR can be increased and account for some of the differences.
Re: Thank you very much for highlighting for point. Accordingly, we have modified the figure (previously figure 2, now figure 3 in the revised version of the manuscript,) to include the quantification by densitometry of total CFTR under the various conditions. Given the variability of these measurements (more evident when comparing total CFTR expression vs processing) the results did not achieve statistical significance.
Minor aspects
Line 45 – It should read “variants” instead of “variant”.
Re: Done
Line 88 – It should read “CRMS” instead of “CMRS”.
Re: Done
Line 108 – Explain why only the ones with F508del were included (sounds obvious but it is better to state it clearly).
Re: We have actually included in our study only the individuals without the F508del variant (and probably this is what the reviewer wanted to highlight). This was done because – in the presence of this variant – it would be difficult to assess the relative contribution of each variant to the total CFTR current measured in patient-derived nasal epithelia. We have clearly stated this in our manuscript as suggested by the reviewer.

Reviewer 2 Report
In this manuscript, Terlizzi et al provide a quite interesting report on the CFTR variant S737F pointing out how collaborative efforts bringing together clinical and translational researchers can provide relevant insight into the effects of rare variants and inform on the best therapeutic options.
There are no major concerns regarding the work presented but some aspects could be improved.
1. The term “variant” is preferred over “mutation”. Please correct the few times in which “mutation” is still used.
The same applies to the term “individuals” which should be used throughout instead of “patients”.
2. The clinical description in the abstract is hard to follow – consider rewriting those sentences to make it more clear.
3. Tables 1 and 2 are difficult to follow – as they include some repeated data. Please consider merging them into a single table, probably in “landscape” format.
4. In figure 1, please consider adding the individual numbers (as they are referred to that in the text) – or instead use only the genotypes in the text.
5. One of the puzzling results is the lack of significant improvement in function after IVA or ETI for the homozygous individual, when compared to the increase observed in individuals in which S737F appears in heterozygosity with a null allele. The authors could probably comment on this in the Discussion and on the possibility that there is a “ceiling” for CFTR activity (?).
6. Another aspect that could be mentioned in the Discussion is the total CFTR expression levels in the different individuals – this is not assessed but could contribute to explaining some of the differences.
7. Finally, the authors could also check, in the western blot data, if total CFTR (and band C) is increased after ETI treatment. Although processing (C over B+C) is not changed, total amount of mature CFTR can be increased and account for some of the differences.
Minor aspects
Line 45 – It should read “variants” instead of “variant”.
Line 88 – It should read “CRMS” instead of “CMRS”.
Line 108 – Explain why only the ones with F508del were included (sounds obvious but it is better to state it clearly).
Author Response

(The authors gave the same response as above.)

Round 2
Reviewer 1 Report
The authors have improved the manuscript a lot by adding more sentences in result and discussion sections and more figures to make data more reliable.
But still in Fig. 1 C, representative traces do not match well with dot plot. In tracing, the drop amplitude of black trace after the addition of inh-172 (IVA) is a little bigger than that of red one (ELX/TEZ/IVA). which is vise versa in dot plot.
Explain in legend Vx-770 marked in traces of figure 1 and 2.
Author Response
AUTHORS’ REPLIES TO REVIEWER’S COMMENTS
REVIEWER #1
The authors have improved the manuscript a lot by adding more sentences in result and discussion sections and more figures to make data more reliable.
We thank the reviewer for his/her appreciation of our efforts to fulfill his/her comments thus improving the manuscript.
But still in Fig. 1 C, representative traces do not match well with dot plot. In tracing, the drop amplitude of black trace after the addition of inh-172 (IVA) is a little bigger than that of red one (ELX/TEZ/IVA). which is vise versa in dot plot.
As stated in the text, and indicated by the statistical analysis, there is no difference in the CFTR activity (determined as the current drop amplitude after the addition of inh-172) detected under the “IVA” condition vs. under the “ELX/TEZ/IVA” condition. Indeed, considering the individual values, there is some overlap (as expected) between the activities detected in the two conditions. The traces that we selected for the graph are – in this view – representative of this “statistically-not-different” activity. We include here the individual values of CFTR activity observed for each epithelium (with the ones corresponding to representative traces highlighted in yellow).
FI062_iva |
FI062_ETI |
12.1951606 |
14.1800475 |
11.7953192 |
13.4844497 |
12.3760412 |
17.0931772 |
11.728679 |
14.1324474 |
14.079983 |
|
14.0891714 |
|
10.4783718 |
Explain in legend Vx-770 marked in traces of figure 1 and 2.
We thank the reviewer for highlighting this missing point. We have substituted the label “VX-770” with “ivacaftor” in all the graphs.
